# Time for Re-Evaluating the Human Carcinogenicity of Ethylenedithiocarbamate Fungicides? A Systematic Review

**DOI:** 10.3390/ijerph19052632

**Published:** 2022-02-24

**Authors:** Pierluigi Cocco

**Affiliations:** Centre for Occupational and Environmental Health, Division of Population Health, University of Manchester, Manchester M13 9PL, UK; pierluigi.cocco@manchester.ac.uk; Tel.: +44-7308-434224

**Keywords:** ethylenedithiocarbamates, Mancozeb, Maneb, Zineb, ethylenthiourea, fungicides, thyroid cancer, melanoma, brain cancer, carcinogenesis

## Abstract

Background. In January 2021, the European Union ended the license of Mancozeb, the bestselling ethylenedithiocarbamate (EBDC) fungicide, because of some properties typical of human carcinogens. This decision contrasts the IARC classification of EBDC fungicides (Group 3, not classifiable as to human carcinogenicity). A systematic review of the scientific literature was conducted to explore the current evidence. Methods. Human and experimental studies of cancer and exposure to EBDC fungicides (Mancozeb, Maneb, Zineb, and others) and ethylene thiourea (ETU), their major metabolite, published in English as of December 2021, were retrieved using PubMed, the list of references of the relevant reports, and grey literature. Results. The epidemiological evidence of EBDC carcinogenicity is inadequate, with two studies each suggesting an association with melanoma and brain cancer and inconsistent findings for thyroid cancer. Experimental animal studies point at thyroid cancer in rats and liver cancer in mice, while multiple organs were affected following the long-term oral administration of Mancozeb. The mechanism of thyroid carcinogenesis in rats has also been shown to occur in humans. Genotoxic effects have been reported. Conclusions. The results of this systematic review suggest inadequate evidence for the carcinogenicity of EBDC fungicides from human studies and sufficient evidence from animal studies, with positive results on three out of ten key characteristics of carcinogens applying to humans as well. An IARC re-evaluation of the human carcinogenicity of EBDC fungicides is warranted.

## 1. Introduction

The European Commission banned the use of Mancozeb, the bestselling ethylene-dithiocarbamate (EBDC) fungicide, from February 2021. Several regulatory agencies from Europe and the United States had previously examined EBDCs and their main metabolite, ethylene thiourea (ETU), with contradictory conclusive statements, based on the same available information. In 2001, the International Agency for Research on Cancer (IARC) Monograph N. 79 downgraded ETU from group 2B to Group 3 of human carcinogens (agent not classifiable as to its carcinogenicity to humans) [1]. Such a decision referred to the industrial uses of ETU as an accelerator for the vulcanization of polychloroprene and other rubbers, to its occurrence as an impurity in preparations of EBDC fungicides, and to its internal exposure as a product of EBDC metabolism. Such decision underwent criticism [2] based on animal studies suggesting a goitrogenic effect and a link with thyroid cancer and a report showing increased TSH levels and genetic damage in agricultural workers exposed to EBDCs [3]. In their reply against the criticism, two members of the IARC Monograph No. 79 Working Group stressed two points [4]:“*The monographs do not formally evaluate the carcinogenicity of metabolites or other endogenously formed substances, although evidence on the biological activity of metabolites may provide important supporting data for evaluating the carcinogenicity of parent substances*”; however, the IARC supplement No.7, in re-evaluating ETU, had extended the Group 3 classification also to the EBDC fungicides Maneb, Thiram, Zineb, and Ziram, without presenting any further evidence [5].“*exceptionally, agents … for which the evidence of carcinogenicity is inadequate in humans but sufficient in experimental animals may be placed in [Group 3] when there is strong evidence that the mechanism of carcinogenicity in experimental animals does not operate in humans*”. In fact, “*…thyroid follicular cell neoplasms… may … be induced by virtually any nongenotoxic goitrogen in these rodent species. In this respect, rodents and humans are quite different…, and no nonradioactive chemical is known to cause these cancers in humans.*”

At that time, only one human study had evaluated the endocrine-disrupting and genotoxic effects of occupational exposure to EBDCs in agricultural settings, using urinary ETU excretion as a biomarker of exposure [3]. The Working Group motivated the decision of discarding it with its small size and the concurrent exposure to different EBDCs and organophosphates. Finally, a lack of evidence that ETU affected thyroid homeostasis in humans was asserted [4].

The EBDC fungicides, such as Mancozeb, Maneb, Zineb, Thiram, and Ziram, are widely used on a variety of agricultural and horticultural crops worldwide (Table 1). Apart from Mancozeb, EBDCs were considered early in the IARC Monographs programme and classified in the IARC Group 3 [5,6,7]. One dithiocarbamate, Disulfiram, is well known for its property to inhibit the enzyme alcohol dehydrogenase [8]. Such property is used in therapy to deter alcoholics from their dependence.

Dithiocarbamates were first synthesized from a monoamine and carbon disulphide, an inorganic solvent, to be used as accelerators in the rubber vulcanization. In 1934, tetramethylthiuram disulphide, known as Thiram, was the first dithiocarbamate to be patented as a fungicide, followed by more active compounds, such as Ferbam and Ziram, especially useful in herbal crops [9]. In 1943, a patent was awarded to the first EBDC, the Hester’s compound disodium ethylene bisdithiocarbamate (Nabam), which rapidly replaced copper sulphate in the treatment of many plant diseases, and became particularly popular among the potato growers in the United States. However, its instability in the solid form complicated its practical handling. Shortly after Nabam introduction in the market, it was discovered that zinc sulphate had a stabilizing effect on the liquid. The reaction product between zinc sulphate and Nabam was zinc ethylene bisdithiocarbamate (Zineb). In 1950, DuPont replaced zinc with manganese and patented manganese ethylene bisdithiocarbamate (Maneb), which was more active than Nabam or Zineb. In 1962, Rohm and Haas registered the zinc ion complex of Maneb (Mancozeb), basically a combination of Maneb and Zineb, the bestseller among the EBDCs [8]. Currently, India accounts for 28% of the Mancozeb global market; 85% of sales are in Pacific Asia and Europe and only 4% in the United States. Further increase in its production and consumption is expected following the development of the Mancozeb production in China, up to an estimate of 250,000 metric tons in 2022 [9,10,11]. Potato and vegetable crops are the major uses (29 and 28% of sales, respectively), followed by orchards (19%) and vineyards (18%). The most important uses in the EU are against early and late blight in tomato and potato crops, the treatment of downy mildews on grapevines and vegetable crops, and the control of scab on pome fruit [9].

The 1996 Extension Toxicology Data Network (EXTOXNET) assessment of Mancozeb described it as practically nontoxic, causing mild skin and eye irritation and sensitization in rabbits and men occupationally exposed. Concern was expressed for chronic exposure because of the occurrence of ETU as an impurity in Mancozeb formulations and its formation in the stored product, capable of causing goitre, birth defects, and cancer in experimental animals [12]. The U.S. Environmental Protection Agency re-evaluated Mancozeb in 2005 [13]. EPA acknowledged a hazard for occupational handlers of the fungicide due to the possibility of skin sensitization and allergy through all routes of administration and to endocrine-disrupting effects at the thyroid level resulting from the ETU metabolite. By extrapolating from the results in experimental animal studies, the cancer risk was defined as low [13]. A September 2008 document of the Swedish Chemicals Agency applied the European Council “cut off” criteria for placing plant protection products in the market to 271 approved active substances. Maneb and Mancozeb were included among the 23 substances meeting such criteria and therefore were ineligible for approval because of endocrine disruption [14]. The U.S. National Institutes of Health National Toxicology Program evaluated ETU in 2010: its conclusions highlighted the sufficient evidence of animal carcinogenicity, with the liver in mice and the thyroid gland in rats as the target organs and the occupational exposure to EBDCs as the main source of intake by inhalation, ingestion, and dermal contact [15]. In the United Kingdom, the Commission Working Group on the Classification and Labelling of Dangerous Substances took Mancozeb under consideration for the first time in November 1993, then again in 2003–2006. In 2017, the Chemicals Regulation Directorate of the U.K. Health and Safety Executive submitted to the European Chemical Agency (ECHA) a proposal for the classification and labelling of Mancozeb [16]. Mancozeb was considered a developmental toxicant. Its neurotoxicity was linked to the blockade of thyroid hormone synthesis, mediated by the inhibition of thyroid peroxidase by ETU, as the thyroid function is crucial for brain development in mammals. Mancozeb was also identified as a cause of allergic skin reaction described as not genotoxic and teratogenic at very high doses, capable of inducing maternal toxicity. Again, the reproductive toxicity at high-level exposure was attributed to ETU, which accounts for 7% Mancozeb metabolic conversion in the experimental animals. As it concerns its carcinogenicity, Mancozeb was defined as a medium-potency thyroid carcinogen in rats, with a non-genotoxic mechanism of action, which would cause thyroid tumours in humans only at implausibly high exposure levels. In absence of clear evidence linking hypothyroidism to thyroid cancer, the U.K. proposal was that Mancozeb classification for carcinogenicity was not required, consistent with the decision taken for ETU [16]. The ECHA opinion was adopted on 15 March 2019 [17]. Evidence that ETU altered thyroid function not only in rats but also in dogs and monkeys at relatively low doses prompted the conclusion that Mancozeb could induce thyroid toxicity in humans at dose levels relevant for classification in Category 2 for thyroid effects. The same was proposed for its neurotoxic effects. No mutagenic effects were detected. As for its carcinogenicity, the inadequacy of the epidemiological studies was highlighted, and the human relevance of the excess of follicular thyroid cancer observed among rats, but not mice, was discussed. The conclusion was that no classification was appropriate with the following arguments [17]:Thyroid tumours in rats arise through inhibition of thyroid peroxidase (TPO) by ETU and/or Mancozeb leading to disruption of the HPT [hypothalamic-pituitary-thyroid] axis, a non-genotoxic mechanism of action.ETU metabolism is more efficient in humans than in rats.The plausible occurrence of the same mechanism of action in humans exhibits large, quantitative differences in respect to adult rats due to their lack of thyroxine-binding globulin (TBG).Thyroid tumours are a relatively common finding in long-term rat studies, whilst the only known human thyroid carcinogen is ionizing radiation.There is no clear evidence of an association between hypothyroidism and thyroid cancer in humans.The epidemiological studies on EBDC exposure and thyroid cancer are negative.A 1999 document by the European Chemicals Bureau (ECB) on thyroid tumours proposed that low- or medium-potency thyroid carcinogens in rodents should not be classified for human carcinogenicity.The Annex VI to the CLP Regulation for ETU, an agent causing thyroid tumours in rats and mice, does not classify ETU as a human carcinogen.

The same year, the European Food and Safety Authority reviewed the ecotoxicological implications of Mancozeb use [18]. The final document acknowledged the developmental toxicity and the endocrine disruption potential of the fungicide and the fact that occupational exposures and bystander exposures were above the acceptable operator exposure level (AOEL) in several crops, including potatoes, cereals, and grapevine. A high risk to birds and mammals from all uses was also highlighted [18]. As a result, on 14 December 2020, the EU State members voted to end the Mancozeb license in January 2021, therefore banning it in the EU from February 2021 [19]. 

This systematic review aims to explore the current evidence on the carcinogenic effects of EBDCs and their derivative ETU to understand whether it might be the time for IARC to re-examine its grading of these strictly linked agents. 

## 2. Materials and Methods

A systematic review of the current status of knowledge about occupational exposure to EBDCs was conducted following the PRISMA guidelines for Systematic Reviews and the related checklist [20]. 

### 2.1. Search Strategy and Study Selection

The exam of the list of references in the 2019 ECHA and 2020 EFSA reports [17,18] resulted in 40 original papers as pertinent to the subject of this review. A further 140 publications through 20 December 2021 were retrieved with PubMed (https://www.ncbi.nlm.nih.gov/pubmed/, accessed on 20 December 2021) using the following search strings: “(ethylenebisdithiocarbamates OR ethylenedithiocarbamates OR Mancozeb OR Maneb OR Zineb OR Thiram OR Ziram OR Metiram OR Ferbam OR Nabam OR Propineb OR ethylenethiourea) AND cancer AND (human OR animal)”, and “(thyroid cancer [title] AND epidemiology AND risk factors AND thyroid function)”, which identified 277 papers. The list of references of each relevant article detected through the automatic search was double-checked to identify further publications. Studies as candidates for inclusion in the systematic review were selected through the title, the abstract, and the text in case of ambiguity. Seventy-nine papers from the first search, and 24 papers from the second were retained. 

### 2.2. Inclusion Criteria

All the studies exploring the different aspects of carcinogenicity of EBDC fungicides, including epidemiological studies, experimental animal studies, and laboratory studies on key characteristics of carcinogens [21], were considered pertinent to this review. Studies related to non-cancer effects were also included because of their plausible relevance to the carcinogenic mechanism at the same target organs.

## 3. Results

### 3.1. Epidemiological Studies on Thyroid Cancer in Agricultural Workers

In 1993, a proportional mortality analysis of death certificates from 23 U.S. states showed an excess from thyroid cancer among farmers [22]. A couple of decades later, this early observation was confirmed among the participants to the U.S. National Cancer Institute Agricultural Health Study (AHS) [23] and in female farmers from five Northern Europe countries [24] but not in the French AGRICAN cohort [25]. A case-control study in Cuba supported the link between a positive story of non-neoplastic thyroid disease and agricultural work and an increased risk of thyroid cancer [26]. A more recent report on occupational thyroid cancer showed a significant increase in risk, particularly for papillary tumours, among pesticide users, with an upward trend by years of employment [27]. A subsequent analysis of the same data set, using a job-exposure matrix, confirmed the increase in risk among women but not men. However, the small number of the exposed prevented any inference to be drawn [28] Consistently, a review of 30 papers on occupational risk of thyroid cancer found a suggestive albeit not conclusive association with agricultural occupations and particularly with use of pesticides [29]. Among the studies examined in that review, two were from the large U.S. National Cancer Institute Agricultural Health Study (AHS), which reported an increase in thyroid cancer incidence in the cohorts of exposed to atrazine and alachlor [30,31]. Another review confirmed the robustness of the association between exposure to insecticides, fungicides, and herbicides and risk of thyroid cancer [32]. Specific agrochemicals of diverse chemical structure, such as the organophosphate malathion [33], the organochlorine chlordane and hexachlorobenzene [34,35], and several chlorophenols, used as disinfectants and insecticides [36], were implicated. 

### 3.2. Epidemiological Studies on Cancer and Exposure to EBDCs 

Table 2 shows details on the epidemiological studies investigating cancer risk associated with exposure to EBDC fungicides.

#### 3.2.1. Thyroid Cancer

The report of an early Russian study of a small cohort of 223 Thiram manufacturers (42 men and 181 women) with more than three years of employment mentioned one case of thyroid cancer and seven cases of enlargement of the gland among 105 workers examined [37]. Another early report from the United Kingdom did not detect an increase in thyroid neoplasms among the female employees of several large rubber-manufacturing plans using ETU and a plant producing ETU [38]. In this regard, the Working Group of the IARC Monograph No.79, which evaluated the human carcinogenicity of ETU, observed that the lack of detail in the description of the methodology and the missing indication of the expected events made it difficult to interpret those findings [1]. Besides, there was uncertainty about whether ETU exposure occurred at the same level in all cohort members.

A mortality study of a small cohort involved in the production of Mancozeb in 1948–1975 at the Rohm and Haas Philadelphia chemical plant did not reveal an excess of cancer deaths at any site in respect to mortality data from the city of Philadelphia. No thyroid cancer death occurred in this cohort [39]. 

An ecological study explored cancer mortality in four areas of the state of Minnesota, with very distinctive geological and agricultural features [40]. The area, where the main crops were wheat, sugar beets, and potatoes, with the most intense use of EBDC fungicides, was characterised by a significant three-fold increase in deaths from thyroid cancer among men with respect to the expected figures, while the risk was only slightly elevated among women [40]. A similar study design was adopted in Sweden, which resulted in the opposite finding [41]. Both studies used indirect information to infer exposure to Mancozeb/EBDC fungicides; however, it is interesting to note that the EU ECHA opinion on the Harmonised Classification and Labelling of Mancozeb [17], initiated by the UK proposal [16], and a recent review of antithyroid drugs and ETU and thyroid cancer [42], mentioned the negative evidence from the Swedish paper but not the Minnesota paper reporting the opposite finding. 

The U.S. AHS provided the ideal setting for conducting a follow-up study on thyroid cancer and pesticide use from the date of enrolment in 1993–1997 through 31 December 2015 [43]. Eighty-five cases of thyroid cancer occurred. Among 44 specific chemicals investigated, significant associations were observed with ever exposure to the fungicide metalaxyl (HR = 2.03, 95% CI 1.16–3.52) and to the insecticide lindane (HR = 1.74, 95% CI 1.06–2.84) but not with ever exposure to Maneb/Mancozeb (HR = 0.51, 95% CI 0.17–1.47), based on five cases only.

#### 3.2.2. Malignant Melanoma

Two studies, one based on the U.S. AHS [44] and another case-control study conducted in one Italian and three Brazilian dermatological hospitals [45], consistently reported an increase in the risk of malignant melanoma of the skin among subjects exposed to Mancozeb. Both studies adjusted the risk estimates by other conditions associated with skin melanoma. These included a tendency to burn, red hair, and average daily exposure to sunlight in the AHS and skin phototype, number of nevi, sunburn episodes in childhood, and family history of skin cancer in the Italian-Brazilian study. Farmworkers spend outdoor a large part of their working hours and therefore are typically exposed to sunlight, the major causal factor for skin melanoma [46]. However, exposure to sunlight was not a risk modifier of the association with pesticide exposure in the U.S. AHS [44], whilst a more than additive interaction was observed when exposure to the fungicide was associated with sunlight exposure [45]. 

#### 3.2.3. Brain Cancer

An excess of brain cancer in association with exposure to Mancozeb was reported in two studies. One, conducted in a neurosurgery department in the Kashmir region of India, observed a strong excess risk among orchard workers using chlorpyrifos, dimethoate, Mancozeb, and captan, with reference to patients with non-malignant brain conditions, such as brain abscesses, meningitis, tuberculoma, multiple sclerosis, and stroke [47]. However, some methodological concerns on the analytical methods and the use of covariates limit the interpretation of those findings. The second study explored brain cancer occurrence in the large French cohort AGRICAN using a Cox Proportional Hazard model for the analysis and assessing exposure with a specific, carefully designed crop-exposure matrix [48]. The results showed an almost two-fold increase in the risk of brain cancer overall, equally shared by the 164 glioma cases and the 134 meningioma cases, in association with EBDC fungicides, including Mancozeb, Maneb, and Metiram, with a significant upward trend by years of use [48].

#### 3.2.4. Other Cancers

Exposure to Mancozeb increased the risk of leukaemia in American farmers [49], and stomach cancer showed no association for ever exposure but a dose-related trend in risk associated with exposure to Mancozeb and Maneb in Hispanic farmworkers of California [50]. Studies of prostate cancer yielded contradictory findings: risk increased with high exposure to Maneb and Mancozeb in a Canadian case-control study [51] but neither the U.S. AHS [52] nor in another population-based U.S. study evaluating environmental exposure using geographic information [53]. The U.S. AHS database was also the setting for a study on colorectal cancer associated with ever exposure to 50 different pesticides, including Maneb/Mancozeb (jointly considered) and Ziram [54]; results were negative for an association with ever exposure to both.

**Table 2 ijerph-19-02632-t002:** Details of the epidemiological studies exploring cancer risk in relation to exposure to EBDC fungicides.

First Author, Year [Ref]	EBDC	Country	Study Design	Exposure	Cancer Site	OR 95% C.I.	Trends	Notes
Cherpak et al., 1971 [37]	Thiram	Russia	cohort	Manufacturing	Thyroid	1 observed case	N/A	
Smith, 1976 [38]	ETU	United Kingdom	Female cohort	Rubber manufacturing; ETU manufacturing	Thyroid	0 observed cases	N/A	
Maher and Defonso, 1986 [39]	Mancozeb	United States	Male cohort	Manufacturing	All cancers Thyroid	0.66 0 observed case	N/A	Unclear exposure definition
Schreinemachers et al., 1999 [40]	EBDC	United States	Ecological	Environmental exposure from agricultural uses	Lip Thyroid	2.7 (1.08–6.71) 2.9 (1.35–6.44)	N/A	
Nordby et al., 2005 [41]	Mancozeb	Sweden	Ecological	Environmental exposure from agricultural uses	Thyroid	0.9 (0.81–1.07) *	No	Risk estimated by the ratio between incidence rates
Lerro et al., 2021 [43]	Maneb/Mancozeb	United States	Cohort	Occupational exposure	Thyroid	0.5 (0.17–1.47)	No	Five cases only
Dennis et al., 2010 [44]	Maneb/Mancozeb	United States	Case-control	Occupational exposure	Skin melanoma	2.4 (1.20–4.90)	Yes	
Fortes et al., 2016 [45]	Fungicides (mainly Maneb/Mancozeb)	Italy, Brazil	Case-control	Occupational exposure	Skin melanoma	3.9 (1.17–12.9)	N/A	
Bhat et al., 2010 [47]	Pesticides (including Mancozeb)	India	Cross sectional	Orchard work	Brain cancer	2.0 (1.86–2.07) **	N/A	
Piel et al., 2019 [48]	Thiram Ferbam, Propineb, Ziram and/or Zineb Maneb/Mancozeb, Metiram	France	Cohort	Occupational exposure	Glioblastoma brain cancer	1.9 (1.09–3.28) 2.2 (1.20–3.67) 1.9 (1.12–3,35)	Yes	
Mills et al., 2005 [49]	Mancozeb Maneb	United States	Case-control	Occupational exposure	Leukaemia	2.4 (1.12–4.95) 1.8 (0.89–3.86)	N/A	
Mills and Yang, 2007 [50]	Mancozeb Maneb	United States	Case-control	Occupational exposure	Stomach cancer	1.2 (0.70–2.06) 0.9 (0.57–1.66)	Yes Yes	
Band et al., 2011 [51]	Ferbam/Maneb Mancozeb Metiram Thiram Zineb Ziram	Canada	Case-control	Occupational exposure	Prostate	1.6 (1.04–2.48) 1.4 (0.89–2.15) 1.4 (0.86–2.42) 0.9 (0.41–2.11) 0.5 (0.22–2.13) 1.5 (0.97–2.27)	Yes No No N/A N/A yes	
Koutros et al., 2011 [52]	Maneb/Mancozeb	United States	Case-control	Occupational exposure	Prostate	0.7 (0.40–1.30)	No	
Cockburn, 2011 [53]	Maneb	United States	Case-control	Environmental exposure	Prostate	0.9 (0.48–1.51)	No	
Lee et al., 2007 [54]	Maneb/Mancozeb Ziram	United States	Cohort	Occupational exposure	Colorectum	0.7 (0.50–1.20) 1.2 (0.50–2.90)	N/A N/A	

Note: * based on the comparison between incidence rates; ** re-calculated based on a cross sectional study design.

### 3.3. Experimental Animal and Laboratory Studies

#### Cancer

Early experimental studies showed that long-term dietary administration of ETU targeted different organs according to the species: thyroid impairment and cancer predominated in male rats at doses of 60 mg/kg; in hamsters, the liver was affected, but doses of 200 mg/kg were not carcinogenic [55]. As mentioned earlier in this paper, besides being the product of the EBDC fungicide metabolism in living organisms, ETU is an industrial chemical used as an accelerator in the rubber industry and an intermediate in the manufacturing of pesticides, dyes, and photographic chemicals. In the past, it was also used as a therapeutic drug for thyrotoxicosis. Therefore, the finding of liver cancer induction in mice following oral administration of ETU raised concern [56]. Thyroid cancer and goitre were reported in rats after feeding them with a 350-ppm dose [57]. A dose-related increase in follicular thyroid cancer manifested starting at 500 ppm in male rats and 1000 ppm in female rats but not in mice, with enlargement of the gland in rats and mice due to hypertrophy and hyperplasia of follicular cells [1]. The carcinogenic effect of ETU on the rat thyroid might be consequent to an increase in DNA damage, as reported in vitro in thyroid cells [58]. Such carcinogenic effect was independent on age at starting the oral administration, but it was slightly higher when it started at perinatal age and continued for two years in the adult age [59]. Because of the metabolic link, the parent EBDC fungicides were also investigated. The 2005 U.S. EPA summarized the studies on Mancozeb and ETU: both caused thyroid cancer in rats following oral administration; in mice, ETU caused thyroid cancer, pituitary adenomas, and liver cancer, while Mancozeb did not cause any type of tumours [13]. However, a long-term (two-year) study of 150 male and female Sprague–Dawley rats fed with Mancozeb at the concentration of 1000, 500, 100, 10, and 0 ppm reported an increase in total malignancies, cancer of the mammary glands, Zymbal gland and ear duct, liver, pancreas, thyroid, osteosarcomas, and lymphoreticular neoplasms, which lead to conclude that Mancozeb was a multipotent carcinogen [60]. In reviewing the experimental animal evidence, the 2017 UK HSE CLH proposal on Mancozeb and the subsequent 2019 ECHA opinion also observed that thyroid hyperplasia of the follicular cells had been shown in rats and dogs at oral doses of 100 ppm [15,16], and while Mancozeb was negative for clastogenicity and aneugenicity in an in-vitro micronucleus study and weakly positive for sister chromatid exchange, it induced in vitro chromosomal aberrations in the presence of dimethyl sulfoxide (DMSO) and micronuclei in rats. Still, both the UK HSE proposal and the ECHA opinion discarded the positive long-term study on rats because of *“… the unusual design of the study and absence of any historical control data, …inconsistency with the findings from the available regulatory carcinogenicity studies on Mancozeb*…[and missing] *information on the purity of the test material*” [16,17]. A quick double-check of the original paper shows that none of those objections was valid: in fact, a control group of 75 rats fed similarly but at 0 ppm of Mancozeb was part of the study; its results were consistent with the U.S. EPA report on Mancozeb and ETU [13], and the purity of Mancozeb was indicated as 85% as the active ingredient [60]. As highlighted in the reports of the regulatory agencies, other cancer sites were positively related to exposure to EBDC fungicides and ETU: adding Thiram and sodium nitrite to the diet of male and female rats resulted in a high incidence of tumours of the nasal cavity [61]. The authors suggested a possible relation with the nitrosamine formation. Another study reported the development of mostly benign skin tumours following the topical application of Mancozeb on the dorsal skin of female Swiss albino mice at a dose of 100 mg/kg body weight dissolved in 100 microliters DMSO three times per week [62]. In a two-stage initiation-promotion study in mice previously painted with a single sub-carcinogenic dose of dimethyl-benz(a)anthracene, the topical application of Mancozeb for 17 weeks caused benign skin tumours in all the treated animals, indicating the fungicide’s action as a promoter [63,64,65]. Such promoting effect might result from increased ornithine decarboxylase activity and DNA synthesis, as indicated by the increase in [3H] thymidine incorporation into the mouse skin DNA [66]. Interestingly, Thiram showed an initiating effect on the skin of Swiss albino mice instead when followed by 12-O-tetradecanoyl phorbol 13-acetate as a promoter [67]. Treatment with ETU and sodium nitrite caused endometrial adenocarcinomas in female mice following hyperplasia of endometrial glands as the precancerous lesion [68] also as a result of N-nitroso ETU formation [69]. Although the highest susceptibility appeared at 6 months of age, the rate of replication of neoplastic endometrial cells was higher in older mice [70]. In another experiment, the same treatment protocol resulted in a dose-dependent increase in the incidence of lymphatic tumours as well as cancer of the lung, forestomach, Harderian gland, and uterus, whilst isolated ETU or sodium nitrite treatment did not show a carcinogenic activity [71].

Mancozeb promoted pancreatic cancer and its progression in rats after nitrosomethylurea (NMU) initiation and in combination with phenobarbital and quercetin but not when administered alone [72,73]. When administered intraperitoneally in pregnant Swiss albino mice, the treatment of the first progeny with the tumour promoter 12-o-tetradecanoyl phorbol-13-acetate (TPA) resulted in a significant increase in tumour incidence in comparison to the progeny of mothers that received only DMSO followed or not by TPA as a promoter. Therefore, Mancozeb or its metabolites may cross the placental barrier, causing DNA damage and tumour initiation in the foetal cells, which would convert to neoplasia following TPA promotion [74]. Propineb also manifested the same promoting effect in male rats after initiation with three nitrosamines on the thyroid, the kidney, and the urinary bladder [75]. In in vitro studies, Mancozeb exhibited properties of a multisite carcinogen, as it induced apoptosis [76,77] through increasing the pool of reactive oxygen species [78] by glutathione depletion due to massive cysteine oxidation [79], a property shared with Thiram [80].

Other EBDCs, such as Thiram and Metiram, did not elicit the appearance of tumours in rodents [81,82]. In the absence of microsomal activation, both Thiram and Mancozeb inhibited thymidine uptake and unscheduled DNA synthesis in a dose-related fashion on in vitro resting and proliferating lymphocytes. In presence of microsomal activation, only Thiram elicited unscheduled DNA synthesis and sister chromatid exchanges significantly more frequently than in controls [83], while Mancozeb and Zineb [84] underwent detoxification. Maneb did prevent the development of neoplasms of the large bowel induced by 1,2-dimethylhydrazine (DMH) in female mice [85], and Mancozeb and Maneb did not show any promoting or co-carcinogenic activity on the rat liver when given in combination with 12 different pesticides [86]. Thiram was also shown to reduce the incidence of leukaemia and pituitary and thyroid tumours [81,87], perhaps because of its powerful inhibition of angiogenesis [88]. Some species, such as newts, were also less sensitive to the carcinogenic action of Maneb [89,90].

### 3.4. Other Health Effects in Experimental Animals and in Humans

#### 3.4.1. Allergy and Contact Dermatitis

The above-mentioned 2017 UK HSE CLH proposal and the subsequent 2019 ECHA opinion on Mancozeb highlighted the development of contact dermatitis in guinea pigs, with cross-sensitization between Maneb and Zineb, and observed that human studies of manufacturing workers had detected cases of contact allergic hypersensitivity [16,17]. Therefore, the H317 (may cause an allergic skin reaction) classification was proposed [16]. Another study, published a few months earlier than the UK HSE CLH proposal, also found an increase in the occurrence of allergic contact dermatitis in Australian Healthcare workers sensitised to haptens in the rubber gloves and specifically to thiuram mix, also containing Thiram [91]. However, no association was observed between EBDC exposure and allergic contact dermatitis, allergic rhinitis, food allergy, and atopy in a previous study [92]. The skin was nonetheless confirmed as a target of EBDC toxicity by the proteomic analysis study on cell culture of human and mouse skin showed that in both cell cultures two proteins, namely calcyclin and calgranulin B, known markers of keratinocyte differentiation and proliferation, were overexpressed [93]. Besides, a report of a case of toxic epidermal necrolysis was described in a man who had applied a dithiocarbamate fungicide in his home garden [94], further pointing at the skin as a sensitive site to the carcinogenic action of EBDC fungicides, and Mancozeb in particular, in humans. 

#### 3.4.2. Neurotoxicological and Neurodevelopmental Effects

Twelve cases of acute intoxication by EBDCs were observed in a French hospital [95]. All the cases exhibited neurological symptoms, such as headache, dizziness, and confusion, and a few of them suffered from seizure episodes, which ceased within a short term. Long-term exposure to pesticides in general and EBDC specifically has been associated with parkinsonism and neurocognitive impairment [95].

In rats treated orally with Mancozeb, dose-related signs of neurotoxicity were also noticed, including adynamia, muscular tone reduction, loss of motor coordination, paralysis of the lower limbs, loss of appetite, and prostration [16,17]. Adverse neurodevelopmental effects were also observed in rats fed with Maneb and Paraquat during puberty, which were prevented by adding taurine to the diet [96]. In vitro studies have shown that Zineb is capable of inducing apoptosis in SH-SY5Y human neuroblastoma cells, suggesting that the same effect would occur in dopaminergic neurons [97]. Additionally, like Paraquat, Maneb increased the level of a-synuclein (a-syn), a neuronal protein critical for normal brain function, and altered the tyrosine hydroxylase and the protein turnover in SH-SY5Y human neuroblastoma cells [98,99]. Ziram was also reported to alter the protein turnover in the same in vitro protocol [100], while Mancozeb did so by inducing changes in the expression of yeast genes related to response to oxidative stress [101]. As a-syn is an amyloid component, these results suggested the involvement of EBDC fungicides in the development of chronic diseases of the central nervous system (CNS), such as Alzheimer’s disease and Parkinson’s disease. These effects would be accompanied by a decrease in mitochondrial respiration, with inhibition of ATP synthesis and an increase in nonmitochondrial respiration. Proton leak would also occur apparently related to the dithiocarbamate functional group, possibly potentiated by manganese [102]. Melatonin seems capable of inhibiting this process [103].

Although a higher plasma β-amyloid level, a pre-clinical feature of Alzheimer’s disease, was associated with an increase in cancer risk [104], a link between the effects of EBDC fungicides at the CNS level and the development of brain cancer is far from being established. Still, the coincidence between the epidemiological studies showing an increase in brain cancer associated with exposure to EBDC fungicides [47,48] and their metabolic effects in in vitro studies is suggestive and worth further exploration.

#### 3.4.3. Genetic Damage and Reproductive Effects

Mancozeb caused dose-related damage to the rat gonads in both genders and, in the presence of signs of maternal toxicity, there was an increase in the frequency of foetal resorption, external haemorrhages, and rib malformations [16]. However, EBDC fungicides and ETU are usually considered not to be genotoxic. The results of previous studies using DMSO or other polar solvents as a vehicle were discarded, as Mancozeb is rapidly degraded and releases metal ions, such as manganese, which can cause genetic damage by themselves [16,17]. However, a paper on Mexican back sprayers using EBDC fungicides described an increase in sister chromatid exchange and chromosomal translocations [3]; no use of DMSO as a vehicle prior to the laboratory analyses was reported in that study. Moreover, in vitro treatment of normal bone marrow mesenchymal stem/stromal cells with a cocktail of seven pesticides, including Maneb and Mancozeb, for 21 days caused inhibition of cell proliferation, DNA damage, and senescence [105]. Additionally, primary bone marrow mesenchymal stem/stromal cells from patients affected by myelodysplastic syndrome could only provide limited support to primitive haematopoiesis [105].

#### 3.4.4. Thyroid Disruption

Thyroid inhibition, with increased TSH levels and decreased T4 levels, was reported in workers exposed to ETU in rubber mixing [106] and agricultural workers in Italy [107] and the Philippines [108] exposed to Mancozeb and other EBDC fungicides. Consistent with the thyroid-inhibiting potential of its ETU metabolite, also used as a biomarker of exposure [3], in a GH3 luciferase gene reporter assay of 21 pesticides, Mancozeb was a potent inhibitor of the thyroid hormone receptor [109]. Mancozeb impairs the hypothalamus-pituitary-thyroid axis in several animal species and affects the weight, volume, and histopathology of the gland. In addition, high-level occupational exposure to Mancozeb is associated with hypothyroidism with increased levels of TSH and hypo-functioning goitre [3,11,110]. Among the exposed to EBDCs in Brazil, TSH blood level increased, and FT4 level decreased with years of exposure [111]. Among the spouses of participants of the U.S. AHS exposed to Maneb/Mancozeb, the risk of suffering from thyroid dysfunction, either in hyper- or hypo-functioning sense, was elevated (OR = 2.2; 95% CI 1.5–3.3) [112,113]. Among the AHS participants suffering from subclinical hypothyroidism, only one pesticide applicator had been exposed to Mancozeb, which prevented any evaluation of the direct exposure in this study [114]. Besides, a significant inverse correlation between urinary ETU excretion and FT4 blood level was observed in 400 pregnant young women from Costa Rica. Twenty-five percent of these women lived within 50 m from a banana plantation, 60% had a partner working at a banana plantation, and 8% worked themselves at a banana plantation while pregnant [115]. As a counterfactual test of the hypothesis, Sheshtra et al. analysed hyperthyroidism among AHS farmers exposed to Maneb/Mancozeb [116]. Their results showed that the risk was halved (HR 0.50, 95% CI 0.30–0.83), which was consistent with an important inhibiting effect of EBDC fungicides at the thyroid receptor level. 

Besides, indirect evidence from ecological studies suggested a higher incidence of thyroid cancer in areas where goitre is endemic and a decline in thyroid cancer incidence following iodine supplementation [117]. Opposite findings have also been published [118] although it is possible that disagreements might be related to differences in histology: in fact, in endemic goitre areas, the follicular and anaplastic forms seem to prevail, while in iodine rich areas, the papillary carcinoma is more prevalent [118]. A 2009 review of the risk factors for thyroid cancer concluded for strong evidence of an association between thyroid cancer and previous adenoma or goitre [119]. A Chinese case-control study provided further support, showing a strong increase in risk of thyroid cancer among subjects who previously suffered from goitre or a benign thyroid disease [120]. More recently, a Serbian case-control study reported an increase in risk of thyroid cancer among subjects who previously suffered from Hashimoto’s thyroiditis and other chronic unspecific thyroiditis [121], further strengthening the existing evidence [122,123,124]. Both a low and a very high iodine intake would affect thyroid function and possibly thyroid cancer [119,125,126] although it is unclear whether a high urinary iodine level would be a cause or an effect of benign thyroid nodules and thyroid cancer [125,127]. The hypothesis has been subsequently confirmed in relation to a mild iodine deficiency in a case-control study [128] and in an ecological study in China [129] and by the decrease in thyroid cancer incidence in strict relation with the decrease in TSH release from the pituitary following the administration of L-thyroxine to patients with a nodular goitre [130]. 

To summarize, strong epidemiological evidence suggests that the inhibitory effects of thyroid function observed in rodents would also occur among humans. As this appears to be the mechanisms of thyroid carcinogenesis in the experimental studies, its observation in humans suggests that EBDC fungicides might cause thyroid cancer also in humans through disruption of the thyroid function [26]. The increase in TSH secretion by the pituitary gland would result in a continuous proliferative stimulus capable of inducing the development of neoplastic foci from cells already carrying initiating mutations.

## 4. Discussion

This systematic review of the existing evidence of human carcinogenicity for EBDC fungicides and ETU highlights differences across molecules, multiple target organs, and research needs. A few studies suggest an excess risk of malignant melanoma and brain cancer among agricultural workers occupationally exposed to EBDC fungicides. In both instances, clinical and experimental evidence of the skin and the central nervous system as target organs for the EBDC toxicity lend credibility to these associations. While the results of the few available epidemiological studies on thyroid cancer are not suggestive for an association, the current evidence is based on two ecological studies reporting contradictory results: one small-size Russian cohort study on rubber workers presumably exposed to Thiram, reporting seven cases of thyroid enlargement and one case of thyroid cancer but no expected events [37]; one short, preliminary report of ETU-exposed workers with no detail whatsoever on exposure level, job title, length of exposure, or duration of follow-up and no indication about the expected events [38]; a third cohort study of researchers of a chemical plant where Mancozeb was also produced, with no reference to specific exposures [39]; and the U.S. AHS follow-up of pesticide use and incidence of thyroid cancer, with only five cases exposed to Mancozeb and/or Maneb [43]. No inference is possible from these studies, as exposure was uncertain or indirectly assumed, or two or more EBDC were jointly considered, no attempts were made to explore trends by surrogates of exposure (such as duration), or the study size was too small. More research is warranted to explore the hypothesis with large, carefully designed epidemiological studies supported by a strong exposure assessment. 

On the other hand, there is sufficient evidence supporting the carcinogenicity of most EBDC fungicides in experimental animals, with the thyroid as the main target in the rats and the liver in mice, with at least one study suggesting a multipotent carcinogenic effect on several other organs [60]. Notably, the EBDC and ETU thyrotoxicity observed in rats, dogs, and monkeys has also been shown in humans, and the link between thyroid disruption and cancer has been well demonstrated in human studies. Further studies are warranted to address the mechanism linking EBDC exposure to thyroid disrupting effects and thyroid cancer in men. It might be important to understand whether any difference in the health effects of specific EBDCs might be related to the release of manganese and zinc ions, consistent with what has been observed in relation to urinary levels of other metals [131,132]. Other mechanisms typical of some but not all EBDC fungicides, such as the oxidative stress and genotoxicity, might also be relevant in human carcinogenesis. 

Despite the existing evidence from experimental animal studies about their potential multisite carcinogenicity, the peculiar link between EBDC fungicides and ETU with thyroid cancer became the focus of some regulatory agencies, while the significance of thyroid tumours observed in rodents was questioned [133]. Rats but not mice are more prone to develop thyroid tumours following chronic stimulation of the thyroid gland by high TSH levels. According to some authors, long-term assumption of drugs that enhance elimination of thyroid hormones would not affect the circulating T3, T4, and TSH levels in humans. As a consequence, “…*non-genotoxic substances that only cause thyroid adenomas/carcinomas in rats, which can be attributed to a disturbance in thyroid function such as the induction of phase II enzymes* …, *are considered of no relevance to humans and do not warrant classification as carcinogenic. This also applies to tumours induced by substances that impair thyroid hormone synthesis or release, such as impaired iodine uptake, inhibition of iodine peroxidase, of thyreoglobulin synthesis, of deiodinases or of hormone release from the thyroid*” [133]. 

Such misinterpretation of findings, upon which the ECHA based its opinion on the classification and labelling of Mancozeb [16,17], and inaccurate accounts of the existing evidence [42] might have contributed to underestimate the toxicological and carcinogenic potential of EBDC fungicides. The current scientific evidence has refuted statements, such as *“there is no clear evidence of an association between hypothyroidism and thyroid cancer in humans*” [16,17] and “*no nonradioactive chemical is known to cause* [thyroid cancer] *in humans”* [133].

Twenty-five years later, it is time to reconsider the objections raised against the IARC decision to classify ETU as a Group 3 human carcinogen and to separate the judgment on this metabolite from the parent substance. The lack of genotoxic potential, whilst at least some of the parent EBDC fungicides do induce genetic damage, might suggest that other derivatives might be implicated. On the other hand, as ETU and EBDC fungicides share the thyroid disruptive potential, that would add to the opportunity of a joint re-evaluation. 

## 5. Conclusions

To summarize the conclusions, it might be profitable to use the IARC scheme as follows:Human studies. The available evidence is inadequate to evaluate the human carcinogenicity of ethylenedithio-carbamates and ethylenethiourea, their main metabolite;Animal studies. There is sufficient evidence of the carcinogenicity of ethylene-dithiocarbamates and ethylenethiourea in experimental animals;Mechanistic evidence. There is sufficient evidence that the mechanisms responsible for the animal carcinogenicity of ethylene-dithiocarbamates and ethylenethiourea also apply to humans.

While the current scientific knowledge fully supports the EU decision to withdraw the license and to ban Mancozeb, it is time for the scientific community to focus on EBDC fungicides with new epidemiological studies supported by state-of the-art retrospective assessment of occupational exposure to EBDC fungicides and sufficient statistical power to detect the association. IARC should consider re-evaluating the potential human carcinogenicity of EBDC fungicides.

## Figures and Tables

**Table 1 ijerph-19-02632-t001:** Structural formula, year of marketing, physical status, and properties of the main ethylene-bis-dithiocarbamates (EBDC).

Dithiocarbamates	Formula	Year Patented	Physical Status	Uses	IARC Last Evaluation (Year, Group)
Disulfiram 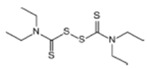		1900	powder	Sulfur vulcanization of rubber; pharmaceutical treatment of alcoholism	1976, 3
Thiram 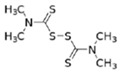	C_6_H_12_N_2_S_4_	1934	powder	Accelerator in rubber vulcanization; pharmaceutical treatment of scabia; sun screen, bactericide; antifungal and animal repellant treatment of seeds, fruit, and ornamental shrubs	1976, 3
Nabam 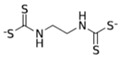	C_4_H_6_N_2_Na_2_S_4_	1943	powder	Antifungal treatment of potato crops and various plants; biocide in sugar millas and pulp and paper mills	Not evaluated
Ferbam 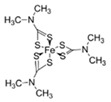	C9H18FeN3S6	1945	Powder, wettable powder, liquid	Fungicide for fruit, nuts, vegetables, ornamental crops, and in household applications.	1976, 3
Zineb 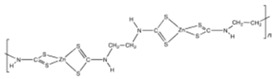	C_4_H_6_N_2_S_4_Zn	1945	Wettable powder	Antifungal treatment of seeds, vegetables, and various field and ornamental plants; additive in paints, fabrics, leather, linen, plastics, and wood surfaces	1976, 3
Ziram 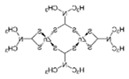	C6H12N2S4Zn	1947	Powder, wettable powder, liquid	Rubber accelerator; fungicide for fruit, vegetables, and ornamental crops.	1976, 3
Maneb 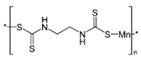	(C_4_H_6_MnN_2_S_4_)_n_	1950	Wettable powder	Fungicide for vegetables, seeds, nuts, field and forage crops, deciduous fruits, grapes, ornamental plants	1976, 3
Metiram 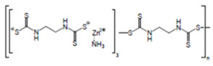	(C_16_H_33_N_11_S_16_Zn_3_)_n_	1958	Wettable powder	Fungicide for cereals, fruits, vegetables, tobacco, and ornamental plants.	Not evaluated
Mancozeb 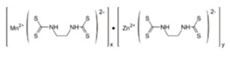	(C4H6MnN2S4)n	1962	Wettable powder	Fungicide for potato, vegetables, orchards, grapes, residential lawn, golf courses, athletic fields	Not evaluated
Propineb 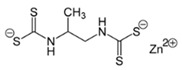	C_5_H_8_N_2_S_4_Zn	1965	Powder, wettable powder, liquid	Fungicide for fruit, grapes, tomatoes, potatoes, tobacco, rice, tea, and ornamental shrubs.	Not evaluated

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
