# Peer review of "Time for Re-Evaluating the Human Carcinogenicity of Ethylenedithiocarbamate Fungicides? A Systematic Review"

_ijerph, 2022, doi:10.3390/ijerph19052632_

Round 1

Reviewer 1 Report

Overall, this is an informative, concise, and well-written review manuscript. The introduction is relevant and theory based. The results are clear and concise.  However, there are some modifications that need to be revised by the author. All comments and suggestions are highlighted in the attached file. So, I recommended publishing this manuscript after revision (Minor Revision).

Detailed comments:

Abstract

What is the novelty of this review?

Lines 9-10: Consider rewriting this sentence.

Line 22: Present a brief conclusion in the end of abstract.

Introduction

Bold the problem associated with this review in the beginning of introduction.

Line 59: Design a table for presenting studied fungicides properties (class, usage, formulation, ...).

Results

You may present some tables or figures to reporting the results.

Regards

Author Response

  1. Abstract. What is the novelty of this review? Lines 9-10: Consider rewriting this sentence. Line 22: Present a brief conclusion in the end of abstract.

In the revised version, The abstract has been modified following the reviewer’s suggestions.

  1. Introduction. Bold the problem associated with this review in the beginning of introduction. Line 59: Design a table for presenting studied fungicides properties (class, usage, formulation, ...).

A new sentence illustrating the problem addressed with this review now starts the introduction. We would avoid using bold characters in the text but, if the editor agrees with the reviewer, we will do it. The revised version includes a table listing the properties and uses of the main dithiocarbamates. All the EBDC fungicides are GHS class 4, and therefore we explained this in the caption, but did not include this information in the Table.

  1. Results. You may present some tables or figures to reporting the results.

A second table summarizing the epidemiological studies on cancer risk associated with exposure to the EBDCs has been added. A table for the experimental studies though would have been too long and difficult to read because of the number of the evaluated outcomes.

Reviewer 2 Report

It is a good review with rather few studies have been performed in both experimental  and epidemiological field for EBDC and their main metabolite ETU. I would prefer to illustrate the sunopsis and comparation of results  and conclutions in one or two separate tables, for better reading and understudying to scientists (countries, experimental, epidemiological results etc).

Although, you analyze all the kinds of toxicity, especially the carcinogecity you have almost nothing refere for the Photo-toxicity, which in my opinion is very important for the farmers and workers. I propose to analyze more the 3.2.2 association with sunlight exposure.

Furthermore, in 3.4.1 Allergy and contact dermatitis, you have not written anywhere, probably you have not found, but you have to refer it, or complete properly.

Author Response

1. “... illustrate the synopsis and comparation of results and conclusions in one or two separate tables”

See the response to the reviewer 1 points 2 and 3.

2. “ … [it] is very important … to analyze … sunlight exposure.”

The paragraph 3.2.2 of exposure. he revised version includes details of the studies on skin melanoma and illustrates the more than additive interaction between EBDC exposure and sunlight in one study and the persistence of the EBDC association after accounting for sunlight exposure.

3. “…Allergy and contact dermatitis … complete properly.”

The paragraph 3.4.1 has not been modified. However, a new sentence in the first paragraph of the discussion now suggests the non-malignant cutaneous effects to indicate the skin as a target organ of EBDC toxicity and possibly carcinogenicity.

Reviewer 3 Report

The review article is well written but there are some comments need to be addressed before the final decision

Major comments:

1- There is no need to divide the review for materials and methods, results and discussion. Please remove this sectioning.

Minor comments:

1- there are some typo errors throughout the review

Author Response

  1. There is no need to divide the review for materials and methods, results and discussion. Please remove this sectioning.

Perhaps the author has missed the point in the guidelines. Anyhow, the sectioning can be easily removed if the editor agrees with the reviewer. It is the author’s opinion that it makes easier the reading.

  1. there are some typo errors throughout the review.

The text has been extensively revised with the aid of an automatic grammar editor. Hopefully, the current version should contain less typo and grammar errors.